# Research Progress on Therapeutic Targeting of Cancer-Associated Fibroblasts to Tackle Treatment-Resistant NSCLC

**DOI:** 10.3390/ph15111411

**Published:** 2022-11-15

**Authors:** Chenxin Li, Yusong Qiu, Yong Zhang

**Affiliations:** Department of Pathology Cancer Hospital of China Medical University, Cancer Hospital of Dalian University of Technology, Liaoning Cancer Hospital and Institute, Shenyang 110042, China

**Keywords:** cancer-associated fibroblasts, non-small cell lung cancer, immunotherapy, tumor microenvironment

## Abstract

Non-small cell lung cancer (NSCLC) accounts for most lung cancer cases and is the leading cause of cancer-related deaths worldwide. Treatment options for lung cancer are no longer limited to surgery, radiotherapy, and chemotherapy, as targeted therapy and immunotherapy offer a new hope for patients. However, drug resistance in chemotherapy and targeted therapy, and the low response rates to immunotherapy remain important challenges. Similar to tumor development, drug resistance occurs because of significant effects exerted by the tumor microenvironment (TME) along with cancer cell mutations. Cancer-associated fibroblasts (CAFs) are a key component of the TME and possess multiple functions, including cross-talking with cancer cells, remodeling of the extracellular matrix (ECM), secretion of various cytokines, and promotion of epithelial-mesenchymal transition, which in turn provide support for the growth, invasion, metastasis, and drug resistance of cancer cells. Therefore, CAFs represent valuable therapeutic targets for lung cancer. Herein, we review the latest progress in the use of CAFs as potential targets and mediators of drug resistance for NSCLC treatment. We explored the role of CAFs on the regulation of the TME and surrounding ECM, with particular emphasis on treatment strategies involving combined CAF targeting within the current framework of cancer treatment.

## 1. Introduction

Lung cancer is a leading cause of death worldwide, accounting for 18% of total cancer-related deaths in 2020, according to global cancer statistics [1]. Approximately 85% of lung cancers are non-small cell lung cancer (NSCLC), of which lung adenocarcinoma (LUAD) and lung squamous cell carcinoma (LUSC) are the most common subtypes. The disease is often diagnosed late, with close to 70% of NSCLC patients having advanced stage disease at diagnosis [2]. Although treatment methods for NSCLC have been continuously developed in recent years, the 5-year survival rate of these patients remains low at 15%. The onset of NSCLC is closely related to multiple types of cancer-associated mutations, of which *EGFR* mutations [3], *ALK/ROS1* rearrangements [4], *MET* exon 14-skipping mutations [5], and *RET* rearrangements [6] have been widely reported. Accordingly, oncogene mutation-targeted therapies have been adopted as the main strategy for lung cancer treatment. The advantages of high effectiveness, low toxicity, and high specificity have made targeted therapy the optimal choice for treating NSCLC patients with driver gene mutations [7]. However, patients have also gradually developed resistance to targeted therapy. For instance, the third-generation epidermal growth factor receptor (EGFR) tyrosine kinase inhibitor (TKI) osimertinib was developed for the treatment of NSCLC patients with the T790M mutation. Although it has demonstrated superiority over first- and second-generation EGFR-TKIs, the development of drug resistance is still a limitation [8]. Platinum-based chemotherapy remains the basic treatment regimen for advanced-stage NSCLC patients without target gene mutations, but the systemic toxic side effects and general drug resistance of chemotherapy remain daunting challenges. Immunotherapy for patients without target gene mutations has led to considerable improvements in progression-free survival and overall survival. However, immunotherapy has its own limitations, with some patients not responding to treatment due to tumor heterogeneity and complexity. Investigations into the behavioral mechanisms of NSCLC are therefore of utmost urgency for the resolution of these issues.

In 1889, Paget proposed the “seed-and-soil” hypothesis in which cancer cells are believed to interact with their surrounding cells, modulating their responses to provide a suitable environment for cancer cell growth and metastasis [9]. This complex ecosystem is now commonly known as the tumor microenvironment (TME) and is mainly composed of tumor cells, cancer-associated fibroblasts (CAFs), and tumor-infiltrating immune cells [10]. It also consists of blood vessels, lymphatic vessels, and the surrounding extracellular matrix (ECM). CAFs are an important component of the TME that can secrete cytokines, growth factors, and CAF-specific proteins and exosomes that will in turn enable ECM remodeling, maintenance of the stemness and metabolism of cancer cells, and regulation of immune status, thereby promoting the proliferation, invasion, metastasis, and drug resistance of tumors [11].

In recent years, although the depth of research on CAFs has increased substantially, a precise definition and specific markers of CAFs are yet to be investigated. It has been reported that CAFs originate not only from the transformation of fibroblasts, but also from the transformation of adipocytes, pericytes, endothelial cells, and mesenchymal stem cells [12]. Currently, it is believed that fibroblasts are quiescent mesenchymal cells embedded in the ECM of interstitial fibers and are activated during wound healing, inflammation, and organ fibrosis [13]. The interaction of fibroblasts and tumor cells with the surrounding cells in the TME causes fibroblasts to exhibit phenotypes with higher activity, which are known as CAFs [14].

An expert consensus statement on CAFs suggested that cells with an elongated morphology that are negative for epithelial, endothelial, and immune cell markers and have not undergone epithelial-mesenchymal transition (EMT) might be considered CAFs [12]. In previous studies, researchers have mainly used α-smooth muscle actin (α-SMA) [13], vimentin (vim) [15], and fibroblast activation protein (FAP) [16] as markers of CAFs. Whether differences between fibroblasts and CAFs are functional or solely indicated by the expression of specific markers remains an obstacle faced in the in-depth investigation of CAFs. Furthermore, the heterogeneity of CAFs is another issue faced by researchers, which may be caused mainly by diversity in the sources of CAFs. Single-cell analysis of CAFs has revealed differences in gene expression, and protein expression and function among different clusters [17]. Recently, using a biobank, prepared with driver genes, of CAFs extracted from biopsies of NSCLC patients, researchers classified CAFS into three subtypes based on resistance to EGFR-TKIs. It has been suggested that HGF, FGF7, and phospho-SMAD2 can be used as functional markers to distinguish CAF subtypes and correlate with targeted therapy and immunotherapy. Among them, subtypes with a high expression of HGF and FGF7 showed strong drug resistance in targeted therapy, and high expression of FGF7 alone also provided some protection. However, only the subtypes with high expression of phospho-SMAD2 showed no protective effect and enhanced CD8 + T infiltration [18]. Although it is generally believed that CAFs mainly exert tumor-promoting effects in tumor onset and progression, certain studies have reported the existence of CAF-related tumor-inhibiting effects. Research on targeting CAFs has also failed to achieve the anticipated therapeutic effects, which may be attributed to the unspecific targeting of these cells. Therefore, improved knowledge on CAF specificity will pave the way for enhanced therapeutic CAF targeting.

Within the current framework of lung cancer treatment, the development of drug resistance in chemotherapy and targeted therapy are inevitable. In addition to gene mutations, the activation of bypass pathways or downstream signaling pathways in cancer cells mediated by and involving CAFs have been proven to be important influencing factors. Given that CAFs regulate the TME through interactions with infiltrating immune cells, their effects in immunotherapy should not be overlooked. A growing number of studies have focused on the roles of CAFs in NSCLC treatment, attempted to resolve the issue of drug resistance, and explored new methods for cancer treatment by influencing CAFs to inhibit NSCLC invasion and metastasis, and promote programmed cell death (PCD) in NSCLC. The present report provides an overview of the new roles and effects of CAFs in NSCLC treatment according to their ability to (1) influence the TME, (2) attenuate the efficacy of existing treatment regimens, and (3) as potential therapeutic targets.

## 2. Regulation of NSCLC by CAFs

### 2.1. CAFs Mediate NSCLC Metastasis and Invasion

CAFs have a dominant role in tumor invasion. Cytokines and/or chemokines derived from CAFs promote EMT of NSCLC cells, supporting them to acquire the mesenchymal phenotype while conferring them metastatic and invasive abilities. Exosomes originating from cancer cells can in turn activate CAFs and enhance their pro-invasive capacity. Hence, interactions between cancer cells and CAFs aggravate NSCLC invasion and metastasis (Figure 1). Furthermore, CAFs can regulate the alignment of collagen fibers and the stiffness of the fibrous stroma; these events promote the morphological and mechanical remodeling of ECM, thereby forming certain tracks or channels in the ECM that can guide the mass migration of cancer cells with an epithelial phenotype [15].

#### 2.1.1. Vascular Cell Adhesion Molecule-1 (VCAM-1)

VCAM-1 is overexpressed in various cancers and is thought to be associated with lung metastasis of breast cancer. Kim et al., based on the GEO database (GSE31210), reported that VCAM-1 is overexpressed in NSCLC compared to that in normal lung tissue and that high VCAM-1 expression in NSCLC is associated with poor survival [19]. In a different GEO database (GSE30219), Zhou et al. also reported that the expression level of VCAM-1 was correlated to the survival rate and metastasis in patients [20]. This study suggested that CAF-derived VCAM-1 activates AKT and MRPK pathways through the VCAM-1/VLA-4 axis. An in vivo xenograft model also showed that VCAM-1 neutralizing antibody treatment blocked the growth-promoting effect of CAFs [20]. VCAM-1 regulation has also received increased attention from researchers. A study showed that CXCL13/CXCR5 axis promotes the migration of lung cancer cells by upregulating VCAM-1 [21]. It was found that miR-26a regulates VCAM-1 levels by inhibiting interleukin-2 signals, which in turn prevents the proliferation and migration of NSCLC cells [22].

#### 2.1.2. C-X-C Motif Chemokine Ligand 12 (CXCL12)

Plerixafor (also known as AMD3100) blocks colony formation in NSCLC induced by CAF-derived CXCL12 [23], and interleukin-24 (IL-24) expression inhibits activation of the protein kinase B (AKT) downstream of the CXCL12/CXCR4 signaling pathway. Therefore, the combined use of IL-24 and plerixafor exerts strong inhibitory effects on NSCLC metastasis [24]. Researchers have also reported that hesperidin inhibits CXCL12 in a dose-dependent manner and significantly reduces the migratory capability of lung cancer cells when used in combination with plerixafor [25].

#### 2.1.3. Matrix Metalloproteinases (MMPs)

MMPs secreted by cancer cells and CAFs are capable of degrading collagen, elastin, and fibrin, thereby promoting cancer cell invasion and metastasis. MMP-2 derived from senescent CAFs can disrupt epithelial adhesion in a transforming growth factor beta (TGF-β)-dependent manner, which strengthens the invasive ability of oral squamous cell carcinoma [26]. In a clinicopathological study, the proportion of CAFs and MMP-9 levels increased gradually during the transformation of adenocarcinoma in situ into invasive adenocarcinoma, with CAFs being positively correlated with MMP-9 expression. Therefore, it is reasonable to believe that CAF activation and increased MMP-9 levels aid LUAD invasion and progression [27]. The up-regulator of cell proliferation [28] and CAF-derived collagen triple helix repeat containing-1 [29] have also been recognized as genes that promote cancer cell invasion induced by MMP-9 overexpression. Moreover, an omega-3-rich microenvironment can suppress the secretion of MMP-9 by CAFs, which is beneficial to downmodulate angiogenesis in the cancer stroma and inhibit cancer cell metastasis [30].

#### 2.1.4. Caveolin-1 (Cav-1)

Cav-1 is one of the main components of caveolae (Ω-shaped invaginations of the plasma membrane). It is expressed in cancer cells, as well as in stromal cells such as fibroblasts, and its abnormal expression in lung cancer is closely associated with cancer progression. A clinicopathological study demonstrated that patients with Cav-1^+^ CAFs had significantly higher rates of vascular and pleural invasion than those with Cav-1^−^ CAFs, and a similar correlation was exhibited by Cav-1^+^ cancer cells [31]. In a phase II clinical trial of combination chemotherapy in lung cancer, patients with high Cav-1 expression in the stroma showed significant improvement in overall survival, but no association was observed regarding Cav-1 expression in cancer tissues [32]. Cav-1-rich CAFs regulate Rho-mediated cell contractility in a p190-dependent manner and promote the local invasiveness and metastasis of cancer cells by remodeling the ECM [33]. However, it is also believed that Cav-1 affects cell motility and invasiveness by triggering the expression of integrins αv and β3 in NSCLC, thereby promoting NSCLC migration [34].

#### 2.1.5. Integrins

Integrins constitute a family of homologous transmembrane cell-matrix adhesion receptors expressed on all nucleated cells. These receptors lack enzymatic activity and recruit cytoplasmic tyrosine kinases upon ligand binding. Various receptor tyrosine kinase signaling pathways participate in the response to integrin activation [35]. Integrin α11 is overexpressed in CAFs and regulates IGF2 expression, thereby promoting the onset and progression of NSCLC [36]. Integrin α11 is also closely associated with LOXL1 expression and plays a key role in collagen crosslinking and stiffness [37]. Similarly, Zeltz et al. demonstrated that integrin α11 regulates LOXL1 expression in CAFs of NSCLC and mediates collagen fiber alignment, thereby supporting tumor growth and metastasis in NSCLC [38].

#### 2.1.6. Growth Arrest-Specific 6 (GAS6)

GAS6 expression in CAFs maintains the protumoral GAS6/AXL paracrine axis and has been described as serving a vital role in the EMT [39,40], drug resistance [41,42,43], and metastasis [44,45] of multiple types of cancer. The human enabled homolog (hMENA), a member of the Ena/VASP family of actin regulatory proteins, modulates inter-cellular adhesion and cell migration. hMENA is highly expressed in CAFs, which is related to collagen contractility. Notably, GAS6 overexpression was detected in cultured CAFs overexpressing hMENA, whereas hMENA silencing decreased GAS6 expression and reduced NSCLC tumor cell invasiveness [45]. Moreover, hMENA/hMENAΔv6 isoforms were also shown to regulate AXL levels in cancer cells, whereas their silencing inhibited GAS6-induced AXL expression and AKT phosphorylation [45].

#### 2.1.7. Platelet-Derived Growth Factors (PDGFs)

Members of the PDGF family participate in CAF recruitment [46] and activation. PDGFs are highly expressed in lung cancer cell lines, whereas CAFs highly express PDGF receptors (PDGFRs). The PDGF/PDGFR axis promotes vascularization, which favors ECM formation and consequently promotes tumor development. Imatinib blocks the transduction of PDGF signals, inhibits CAF proliferation, and reduces the stimulatory effects of CAFs on cancer cells [47]. Cancer cells that have undergone EMT secrete PDGF-BB, which aids in the enhancement of the invasive ability of CAFs and provides physical tracks for cancer cell invasion. Results of an in vitro experiment revealed that downregulation of PDGF-BB or use of a PGDFR inhibitor (imatinib) reduce the invasive ability of CAFs and cancer cells [48]. Furthermore, the PGDFR inhibitor crenolanib is also capable of blocking PDGF signals in a dose-dependent manner, thereby decreasing NSCLC cell viability and promoting cancer cell apoptosis [49]. However, the effects of crenolanib on CAFs have not been comprehensive explored to date.

### 2.2. CAFs Regulate the Immune Microenvironment of NSCLC

As discussed above, the crosstalk between CAFs and tumor cells plays an important role in the occurrence and development of NSCLC. The crosstalk between CAFs and immune cells plays a key role in shaping the immune microenvironment. Previous studies have demonstrated that CAFs induce immunosuppressive cells including M2 macrophages [50], regulatory T cells [50], and myeloid-derived suppressor cells (MDSCs) [51]. CAFs can inhibit CD8+ T cell functions [52]. In addition, lymphocytes can induce PD-L1 expression in CAFs [53], further promoting the immunosuppressive microenvironment. In conclusion, CAFs can affect the expression of the programmed death ligand-1 (PD-L1), modulate immune cell recruitment, and participate in ECM remodeling through the secretion of TGF-β, chemokines, cytokines, and various ECM components, which contribute to the immune regulation of the TME (Figure 2).

#### 2.2.1. NADPH Oxidases

The transdifferentiation of fibroblasts into CAFs is dependent on the delayed phase of intracellular ROS production by NADPH oxidase 4 (NOX4), and this transdifferentiation may be independent of TGF-β signaling to promote tumor progression [54]. NOX4 accumulation in CAFs may help inhibit infiltration of CD8+T cells [52]. The NOX4 inhibitor, GKT137831, inhibits CAF differentiation, reduces CAF levels, and increases CD8+T cell infiltration. The combination of GKT137831 and PD-1 checkpoint inhibitors also showed better efficacy. In LUSC, CAFs are positively associated with the abundance of monocytes, and induce the polarization of monocytes into the MDSC phenotype. The accumulation of MDSCs produced more NOX2 and excluded CD8+T cell infiltration. NOX2 inhibitors and ROS scavengers also significantly reduced the inhibition effect of TME on CD8+T cells [51]. In conclusion, NADPH oxidase plays an important role in both the production of CAFs and the promotion of immunosuppressive TME through the secretion of chemokines by CAFs. The inhibition of NADPH oxidase can also be explored as a great therapeutic alternative.

#### 2.2.2. Programmed Death-1 (PD-1)/PD-L1 Axis

PD-1 is an inhibitory receptor expressed by T cells, and its main ligand PD-L1 is expressed in cancer cells and surrounding stromal cells. PD-1/PD-L1 interactions mediate immunosuppression in the TME, and PD-1/PD-L1 inhibitors have already been widely used in advanced NSCLC patients. Being the major component cells of the TME, CAFs may potentially serve as targets to counter drug resistance to PD-1/PD-L1 inhibitors. Various CAF-derived chemokines and cytokines recruit PD-1^+^ lymphocytes to form the immunosuppressive microenvironment, and CAFs are also capable of inducing PD-L1 expression in cancer cells and surrounding stromal cells. Indeed, LUAD cells were shown to overexpress PD-L1 in response to the stimulus of CAFs-conditioned medium, which was believed to be partly mediated by CXCL2 [55]. In recent years, considerable attention has been paid to PD-L1 testing in cancer cells, whereas certain researchers have discovered that PD-L1 expression on CAFs also partakes in drug resistance in immunotherapy. PD-L1 expression on CAFs is regulated by interferon gamma (IFN-γ) released from cytotoxic T lymphocytes and is an independent prognostic factor of longer postoperative relapse-free survival [53].

#### 2.2.3. Adenosine

In the TME of NSCLC, the ectonucleotidases, CD39 and CD73, serve the main purpose of converting extracellular ATP into adenosine. Studies have demonstrated that adenosine possesses immunosuppressive properties and induces the formation of the tumor immunosuppressive microenvironment by binding to different adenosine receptors [56]. CAF-derived CD39 mediates PD-1^+^ lymphocyte infiltration, and CD73 is associated to the regulation of FOXP3^+^ lymphocyte infiltration [57]. CAFs not only overexpress CD39, but also induce CD39 expression on T cells, and activated T cells can in turn promote CD73 expression in CAFs [58]. These findings suggest synergistic effects between T cells and CAFs in adenosine generation.

#### 2.2.4. CXCL12

The chemokine CXCL12 is mainly expressed in the TME through CAFs, mediating at least partly their immunosuppressive effects. Moreover, binding of CXCL12 to the C-X-C chemokine receptor type 4 receptor impairs the infiltration of T and natural killer cells. Zboralski et al. demonstrated that CXCL12 inhibition enhances the infiltration of CD8^+^ and CD4^+^ T cells, as well as natural killer cells in H1299 cell line-derived tumors. In vivo findings also showed that the combination of NOX-A12 (a CXCL12 inhibitor) and anti-PD-1 therapy improved the antitumor effects of immunotherapy [59].

#### 2.2.5. Interleukin-6 (IL-6)

IL-6 is a chemokine that is mainly produced in tumors in an autocrine manner by tumor cells and CAFs. IL-6 induces MDSC recruitment and macrophage polarization and can serve as a biomarker of the suppression of CD8^+^ T cell infiltration in NSCLC, which is known to contribute to the suppression of the host immunity [60]. In a xenograft model of adenocarcinoma, the anti-IL-6 antibody siltuximab exhibited more potent effects in tumor cells in the presence of CAFs than without these cells [61]. CAF-derived IL-6 has a crucial role in STAT3 activation in neutrophils, activating PD-L1^+^ neutrophils and impairing T cell function via the PD-1/PD-L1 pathway in hepatocellular carcinoma [62]. Therefore, IL-6 can contribute to the formation of the tumor immunosuppressive microenvironment by regulating various types of immune cells, such as T cells and MDSCs.

#### 2.2.6. Podoplanin (PDPN)

PDPN, a glycoprotein also known as a marker for lymphatic vessels, is a functional molecule expressed by CAFs. Studies have shown that PDPN^+^ CAFs are closely associated with poor outcomes in NSCLC [50,63,64,65,66,67]. In LUAD, the presence of PDPN^+^ CAFs is recognized as a predictive marker of shorter progression-free survival, but not with shorter overall survival, in recurrent patients who received platinum-based chemotherapy [65]. PDPN-overexpressing CAFs are associated with multiple immunosuppression-related cytokines. In stage I LUSC, the number of stromal CD204^+^ tumor-associated macrophages (TAMs) was reported to be significantly higher in patients with PDPN^+^ CAFs than in those with PDPN^−^ CAFs [50]. These findings have been corroborated by another study on patients with stage I LUAD, which reported a significant correlation between CD204^+^ TAM levels and PDPN^+^ CAFs [68]. Although the number of PDPN^+^ CAFs and FOXP3^+^ tumor-infiltrating lymphocytes are correlated in LUAD, a correlation has not been observed in LUSC, and no associations have been found between CD8^+^ T cells and PDPN^+^ CAFs in adenosquamous carcinoma. The above described results indicate that PDPN^+^ CAFs promote the formation of the immunosuppressive microenvironment in NSCLC and may be associated with resistance to EGFR-TKIs [69]

#### 2.2.7. Immune-Adjuvant Effects of CAFs

Besides contributing to the immunosuppressive TME, CAFs also reportedly exert certain immune-adjuvant effects. In a survival analysis of NSCLC patients, FAP+ CAFs in CD3/CD8-enriched tumors were found to be associated with enhanced patient survival [70]. However, PDGFRβ was an independent negative marker of survival in patients with low expression of CD8+T. Therefore, the immunomodulatory effects of different CAFs markers are not the same or even contradictory, which strongly suggests that there is strong heterogeneity among CAFs. In conclusion, we believe that in addition to considering the heterogeneity of CAFs itself, different immune microenvironments caused by the enrichment and deficiency of immune cells are also important influencing factors. In addition, we noted that the authors included a large number of NSCLC cases in this study. Unfortunately, the authors did not classify the two histological subtypes, LUAD and LUSC. We believe that because LUAD and LUSC have varied tumor development and molecular basis, it is necessary to classify histological subtypes in TME correlation analysis. A previous study by this team has also confirmed our idea that FAP+CAFs is associated with improved prognosis in patients with LUSC, but not in patients with LUAD [71].

Moreover, CAF-derived CXCL12 and IL-6 are believed to participate in the recruitment of cytotoxic immune cells [72]. The conflicting effects of CXCL12 and IL-6 have been described in detail in a review of TME. IL-6 can stimulate the exosmosis of CD8+T cells; in addition, IL-6 promotes the accumulation and activation of MSDC and plays a role in the polarization of macrophages and neutrophils towards immunophenotypes. Similar to IL-6, CXCL12 is also involved in recruiting immunosuppressive cells, although it recruits NK cells and CD8+T cells. Therefore, we believe that CAFs are more likely to promote the immunosuppressive TME to play the pro-tumor function.

### 2.3. CAFs and Cell Death

Cell death is regulated by various complex intracellular and extracellular signals, and the imbalance between proliferation and death of cancer cells represents the foundation for cancer onset and progression. In recent decades, our understanding of the PCD pathways of has become increasingly comprehensive. Apoptosis, the main form of PCD, can be triggered by intrinsic mitochondrial pathways (e.g., the BCL-2 pathway) or extrinsic death receptor pathways. Other major forms of PCD include necroptosis, autophagy, ferroptosis, and pyroptosis. In addition to the effects exerted by CAFs on the PCD and necrosis of NSCLC cells, PCD of CAFs also plays a key role in tumor progression.

#### 2.3.1. PCD of CAFs

##### Autophagy

Autophagy serves as an intracellular degradation system by which cytoplasmic materials are delivered into lysosomes for degradation to maintain intracellular homeostasis. CAFs exhibit increased autophagy compared with normal fibroblasts [73]. The activation of CAF autophagy promotes the proliferative and invasive activity of CAFs towards lung cancer cells, and a significant upregulation of EMT and metastasis-related genes also occurs in NSCLC cells. The autophagy-dependent secretion of HMGB1 by CAFs activates the NF-κB signaling pathway in lung cancer cells, which in turn enhances the invasive and metastatic activity of these cells [73]. A study showed that normal human lung fibroblasts treated with cigarette smoke extract overexpress the autophagy-related proteins optineurin and Rab1B, and subsequently secret high levels of interleukin-8 (IL-8), thereby promoting cancer cell invasion. In addition, pre-treatment with the autophagy inhibitor chloroquine significantly reduced the ability of cigarette smoke extract to promote cancer cell invasion [74]. Interestingly, Hupfer et al. reported that a mere increase in matrix stiffness was sufficient to increase the autophagy levels in normal fibroblasts and CAFs [75]. Another study showed that p62 levels are lower in CAFs than in normal tissues in early-stage cancer. Specifically, p62 is transcriptionally induced and rapidly degraded via autophagy in CAFs derived from early-stage cancers, which in turn induces the activation of CAFs. In addition, normal human lung fibroblasts lacking p62 and subjected to hypoxic induction or co-cultured with A549 cells showed a significant decrease in the expression of LC3B, an autophagic flux marker. The expression of CAF markers was also significantly downregulated, which inhibited initial tumor progression [76].

Regardless of the autophagy-dependent differentiation of normal fibroblasts into CAFs or the provision of suitable microenvironments for cancer cells by CAFs through autophagy, the intracellular degradation system established by autophagy occupies a pivotal position in cancer onset and development. Hence, targeting autophagy-related signaling pathways may serve as a novel approach and direction for cancer treatment.

##### Necroptosis

Necroptosis is a form of PCD that occurs downstream of the receptor-interacting protein kinases RIPK1 and RIPK3. Specific activation of RIPK1/RIPK3 within fibroblasts can drive antigen uptake and the activation of tumor antigen-presenting cells. These events induce beneficial inflammatory responses and potentiates the antitumor immunity of CD8^+^ T cells, which may lead to long-lasting tumor rejection when co-administered with anti-PD-1 therapies [77].

#### 2.3.2. PCD of NSCLC Cells

##### Apoptosis

An increasing number of studies have focused on the intrinsic and extrinsic apoptotic pathways of cancer cells [78], expecting to identify new avenues for antitumor therapy through the regulation of cancer cell apoptosis. Wang et al. reported that CAF-derived exosomal miR-103a-3p effectively downregulates the expression of the pro-apoptotic protein Bak1 in H1299 cells, consequently protecting them from apoptosis [79]. Sun et al. reported that the upregulation of the long non-coding RNA (lncRNA) HOTAIR in CAFs induced the secretion of the anti-apoptotic protein Bcl-2, thereby inhibiting the apoptosis of NSCLC cells [80]. The CAF-derived lncRNA ANRIL has also been shown to increase Bcl-2 expression [81]. Brünker et al. designed and engineered a bispecific FAP-DR5 antibody that simultaneously targeted the FAP protein in CAFs and the death receptor 5 (DR5) in tumor cells, thereby providing an extrinsic apoptotic pathway that was targeted towards DR5 clustering and capable of specifically activating tumor cells [82]. Results of this study also demonstrated that FAP-drozitumab promoted NSCLC cell apoptosis in a dose-dependent manner.

##### Pyroptosis

Pyroptosis is an inflammatory form of PCD characterized by cell swelling, cell lysis, and the release and activation of proinflammatory cytokines. When GSDMD, the key effector of pyroptosis, is cleaved by caspase-1, its N-terminal fragment forms pores in the plasma membrane, thereby allowing the release of bioactive interleukin-1β, interleukin-18, and other cell contents. Indeed, researchers developed a pyroptosis-related prognostic risk score (Pyro score) for LUAD based on pyroptosis-related genes. Patients with high Pyro scores showed poor prognosis and significant enrichment of stromal-related signaling pathways; in particular, TGF-β was associated with stromal activation in the high Pyro score group [83]. Unfortunately, the study did not elaborate on the presence or absence of relevant relationships between the Pyro score and CAF-related markers, but is reasonable to speculate that TGF-β, which acts as a CAF activator, may affect pyroptosis through its effects on CAFs.

##### Ferroptosis

In the typical regulatory pathway of ferroptosis, cystine enters cells through the cystine/glutamate antiporter system with concomitant accumulation of glutathione (GSH). With the aid of GSH, glutathione peroxidase 4 (GPx4) promotes the reduction of phospholipid hydroperoxides to their respective alcohols. Thus, a decrease in GPx4 levels leads to the accumulation of phospholipid hydroperoxides, which causes damage to cell membrane structures and ultimately results in ferroptosis. In other words, ferroptosis is driven by lethal lipid peroxides, and is also a result of cell metabolism and oxidation-reduction imbalance [84]. FAP overexpression may be positively correlated with GPx4, which consequently inhibits cancer cell ferroptosis [85]. Indeed, CAF-derived gamma-glutamyl-transferase 5 (GGT5) promotes GSH transfer from the TME to LUAD cells, thereby reducing intracellular ROS levels and inhibiting cancer cell apoptosis. Given that FAP overexpression may be associated to the inhibition of ferroptosis in cancer cells, we speculate that FAP^+^ CAFs may inhibit ferroptosis through the effects of GGT5 on GSH and ROS levels, but no related reports have been published. Hu et al. developed *FAP*-engineered tumor cell-derived exosome-like nanovesicles (eNVs-FAP) as a tumor vaccine that can be easily prepared. In addition to upregulating IFN-γ expression and attenuating the tumor immunosuppressive microenvironment, the eNVs-FAP vaccine also depletes FAP^+^ CAFs and simultaneously induces ferroptosis in tumor cells via two different pathways [86]. A recent study indicated that the CAF-derived miR-522 inhibits ferroptosis in cancer cells by targeting the arachidonate 15-lipoxygenase [87]. This above described evidence clearly demonstrates the existence of a correlation between FAP expression and ferroptosis in cancer cells; however, the intrinsic mechanisms of the relevant interactions remain unclear and require further research for elucidation.

## 3. Roles of CAFs in NSCLC Treatment

### 3.1. CAFs Mediate Resistance to TKIs

NSCLC tumors harboring *EGFR* mutations are sensitive to TKIs, with responses observed in approximately 60–70% of patients [88]. Unfortunately, drug resistance inevitably develops and causes disease progression. Recent studies suggest that EGFR-TKI-resistant cancer cells undergo EMT [89]. Interestingly, CAF subpopulations isolated from EGFR-TKI-resistant tumors can induce EMT and inhibit the EGFR-TKI-mediated blockade of the EGFR pathway in co-cultured tumor cells [90] (Table 1).

#### 3.1.1. HGF/MET Pathway

Hepatocyte growth factor (HGF) is another important CAF-derived cytokine. It has been demonstrated that NSCLC cells and CAFs influence each other through the HGF/MET pathway. Kanaji et al. reported that HGF secreted by CAFs promotes cancer cell survival in vivo, whereas inhibition of MET suppressed HGF-induced tumor progression [91]. MET amplification and the activation of its ligand HGF are widely reported bypass pathways of EGFR-TKIs resistance. Indeed, CAF-derived HGF was shown to restore the PI3K/AKT signals by activating MET regardless of the EGFR and ErbB3 activation status, and thus, effectively inducing gefitinib resistance of lung cancer cells with EGFR-activating mutations. The ability of anti-HGF antibodies and MET-TKIs to reverse the resistance of NSCLC to gefitinib has also been validated in a preclinical study [92]. Moreover, it was shown that the MET inhibitor capmatinib blocks the MET/AKT/snail signals in vivo, and that the capmatinib and osimertinib combination not only overcomes drug resistance by targeting both tumor cells and CAFs, but also results in a decreased ability of cancer cells to form tumor colonies, migrate, and generate tumor spheres [93].

#### 3.1.2. FGF/FGFR Pathway

Fibroblast growth factor (FGF) and its receptor FGFR regulate cell migration, proliferation, and metabolism through complex signal transduction pathways. Overexpression of FGF2 in CAFs and FGFR in tumor cells promoted NSCLC progression, whereas the FGFR inhibitor AZD4547 significantly reduced the size and number of tumor nodules in an in vivo xenograft model [94]. Strong synergistic effects were observed during combination treatment with the FGFR1 inhibitor and gefitinib, and the sensitivity of drug-resistant cells was restored by the FGFR1 inhibitor [95,96]. More importantly, a clear mesenchymal phenotype was exhibited by resistant EGFR mutant NSCLC cells, with FGF2 and FGFR1 expression increased in these cells. Combined FGFR inhibitor (BGJ398) and gefitinib treatment provided significant growth suppression effects, with the synergistic effects mainly manifested in the mesenchymal models of NSCLC but not in the epithelial models [97].

#### 3.1.3. GAS6/AXL Pathway

GAS6, which is mainly produced by CAFs, serves as a ligand for AXL and promotes the invasiveness of gastric cancer cells via AXL activation [44]. AXL, which is considered a characteristic marker of EMT, is a receptor tyrosine kinase proven to be overexpressed in mesenchymal-type NSCLC, with its levels being positively correlated with vim [40]. Moreover, patients with mesenchymal-type NSCLC tumors are more prone to develop resistance to EGFR-targeted therapy and exhibit higher levels of AXL. In contrast, treatment with the AXL inhibitor SGI-7079 was shown to reverse resistance to the EGFR inhibitor erlotinib and effectively inhibit tumor growth in vivo [42]. Similarly, in the absence of the *EGFR* T790M mutation or MET activation, AXL expression was shown to be upregulated and to mediate the EMT of resistant cells [43]. These above described findings indicate that AXL activation mediated by interactions between CAFs and cancer cells may serve as a critical target for counteract resistance to EGFR-TKIs.

#### 3.1.4. OSM/OSMR Pathway

CAFs secrete various cytokines and interact with cancer cells in a paracrine fashion, thereby influencing NSCLC progression and mediating resistance to targeted therapies. Researchers found that the CAF-derived IL-6 family cytokine oncostatin-M (OSM) induced a switch to the EMT phenotype via paracrine activation of the OSM receptors (OSMRs)/JAK1/STAT3 axis, thereby protecting cells from targeted drug-induced apoptosis. Combination treatment using the selective JAK1 inhibitor filgotinib and targeted therapy was shown to reduce resistance to targeted drugs. Interestingly, downregulation of OSMR expression was observed after JAK1 knockdown. These findings suggests the presence of feedback activation of OSMR by JAK1, which form a vicious cycle in drug-resistant cells [40].

#### 3.1.5. MEK/ERK Pathway

Resistance to EGFR-TKIs in lung cancer cells is not merely activated through bypass signaling channels but may also be mediated via the amplification of downstream signaling pathways. In an in vitro model, resistant CAFs derived from drug-resistant NSCLC cells exhibited an increase in MEK/ERK expression. More importantly, compared with the sensitive CAFs, resistant CAFs secreted more inflammatory cytokines, such as IL-6, IL-8 and HGF. Resistant CAFs also promoted the development of osimertinib resistance in osimertinib-sensitive P_AZDS1 and H1975 cells [98]. Hence, drug-resistant NSCLC induce the generation of CAFs with higher carcinogenicity, which in turn promote the development of osimertinib resistance. Furthermore, MEK/ERK levels in CAFs and NSCLC cells are closely associated with osimertinib resistance. Indeed, combined use of the MEK inhibitor trametinib and osimertinib significantly reduced the viability of NSCLC and resistant CAF cells, thereby inhibiting tumor growth [98]. When the NSCLC adenocarcinoma cell line PC-9 was co-cultured with PDPN^+^ CAFs, the cancer cells exhibited strong gefitinib resistance with concomitant elevation of ERK phosphorylation levels [69].

### 3.2. CAFs Mediate Resistance to Chemotherapy

For advanced lung cancer patients who are not recommended for surgery, chemotherapy is the most common and possibly the only choice of treatment. Therefore, the decrease in survival rates caused by resistance to chemotherapy is a problem that needs to be urgently addressed. It is generally recognized that CAFs engage in crosstalk with neighboring cancer cells, thereby regulating the biological behaviors of cancer cells and promoting cancer progression [12]. Hypoxia induces glycolysis in drug-resistant NSCLC cells, resulting in increased expression of the key glycolytic enzyme pyruvate kinase isozyme type M2 (PKM2). In turn, exosomal PKM2 can be delivered from drug-resistant cells to CAFs, which promote their metabolic reprogramming and induce cisplatin resistance in A549 cells [99]. The effects of CAFs on cisplatin resistance in NSCLC have also received considerable research attention (Table 2).

#### 3.2.1. IL-6/IL-6R Pathway

Anti-IL-6 receptor (IL-6R) antibodies can partly attenuate changes in the EMT phenotype induced by CAFs and also inhibit CAF-mediated cisplatin resistance. IL-6 secreted from CAFs can exert synergistic effects along with TGF-β signaling and form a vicious cycle between cancer cells and CAFs. Immunohistochemical data have demonstrated that stromal IL-6 expression is significantly correlated with EMT marker expression, tumoral TGF-β expression, and distribution of α-SMA staining. In addition, stromal IL-6 expression was also significantly correlated with poorer patient prognosis and served as an independent prognostic factor for NSCLC [100]. The anti-IL-6R monoclonal antibody tocilizumab was shown to reverse the effects of fluorouracil resistance and significantly reduce tumor weight in a xenograft mouse model of gastric cancer [107]. Researchers also found that the natural-based antitumor compound T21 was capable of inhibiting STAT3 activation induced by IL-6 stimulation and that reduced survivin levels may aid in overcoming IL-6-mediated resistance of NSCLC to chemotherapy [101]. In addition to IL-6, interleukin-11 (IL-11) has also been reported as being capable of upregulating the expression of the anti-apoptotic protein Bcl-2 and survivin in cancer cells by activating the STAT3 downstream signals through its receptor IL-11R. Therefore, IL-11 can strengthen the anti-apoptotic ability of LUAD cells subjected to cisplatin therapy and induce cisplatin resistance [102].

#### 3.2.2. IGF/IGFR Pathway

The insulin-like growth factor 2 (IGF2) is another key growth factor secreted by CAFs, which is believed to be promoted by the activation of integrin α11 in CAFs [36]. IGF1 and HGF can stimulate IGFR- and MET-mediated annexin 2 (ANXA2) expression and phosphorylation, thereby promoting EMT in NSCLC [108]. In an in vitro culture experiment, CAF-derived IGF2 was shown to activate the AKT/Sox2 pathway by binding the membrane receptor IGF-1R in NSCLC cells, which consequently leads to the regulation of drug resistance in these cells. Moreover, the IGF-1R inhibitor OSI-906 was shown to block A549 drug resistance in vivo, which significantly reduced tumor volume and prolonged survival time [103].

#### 3.2.3. Collagen/Integrins

Chemoresistant NSCLC cells typically exist as quiescent-like, slow-cycling cancer cells, which recruit CAFs in a paracrine manner. Upon activation, CAFs secrete collagen, which transmits cell proliferation signals via integrins. Treatment with the Src inhibitor dasatinib was shown to attenuate cell proliferation and colony formation of drug-resistant cells and promote apoptosis. Noteworthily, immunohistochemical analysis revealed that combination therapy with dasatinib and celecoxib significantly inhibited the growth of paclitaxel-treated NSCLC adenocarcinoma in vivo [104].

#### 3.2.4. CCL5/CCL5R Pathway

Primary CAFs isolated from NSCLC patients exhibit increased expression of C-C motif chemokine ligand 5 (CCL5), which in turn upregulates the expression of the lncRNA HOTAIR in A549 cells. This mechanisms was shown to partial inhibit cisplatin sensitivity of NSCLC cells, thereby attenuating cisplatin-induced apoptosis [80].

#### 3.2.5. GAS6/AXL Pathway

In addition to the aforementioned possibility that the GAS6/AXL axis mediates resistance to EGFR-TKIs, researchers have also discovered CAFs of the Lewis cell line and NSCLC patients who had been treated with chemotherapy overexpress GAS6. Moreover, the 5-year disease-free survival rates of patients with AXL^+^ tumors and GAS6^+^ stromal cells are significantly reduced [41].

#### 3.2.6. Gamma-Glutamyl Transferase (GGT)

GGT is a membrane-dependent enzyme that participates in the extracellular degradation and metabolism of the antioxidant GSH. High serum GGT levels have been proven to be an independent prognostic factor for short progression-free survival and overall survival in NSCLC patients, which may possibly be attributed to the induction of resistance to platinum-based chemotherapy by GGT [109]. Furthermore, the mRNA levels of *GGT5* were found to be correlated with overall survival and progression-free survival of LUAD patients. Indeed, GGT^+^ cells were mainly detected in the tumor stroma and were overexpressed by CAFs. CAF-derived GGT5 was shown to promote the transfer of GSH from the TME to LUAD cells, thereby reducing their intracellular ROS level and consequently promoting the inhibition of cancer cell apoptosis and cisplatin resistance [105].

#### 3.2.7. Annexin A3 (ANXA3)

ANXA3 is a phospholipid- and membrane-binding protein regulated by calcium ions that is believed to participate in chemotherapy resistance in hepatocellular carcinoma [63]. Notably, compared with normal fibroblasts, CAFs have higher expression of ANXA3. It has been found that CAFs can promote the activation of the ANXA3/JNK pathway in A549 and H661 cells, which in turn enhances cisplatin resistance in both cell lines. The upregulation of survivin expression also suggests that CAFs inhibit cisplatin-induced apoptosis [106].

### 3.3. CAFs Regulate Responsiveness to Immunotherapy

Considering that CAFs participate in the regulation of immune microenvironments, certain researchers assert that CAFs are capable of regulating the responsiveness of cells to immunotherapy. The colony stimulating factor 1 (CSF1) is a hematopoietic growth factor involved in the proliferation, differentiation, and survival of monocytes, macrophages and bone marrow progenitor cells. It is also a key regulator of TAM differentiation and survival. Although TAM exhaustion has been observed with the use of the CSF1R inhibitor JNJ-40346527 in various tumor models, satisfactory antitumor effects have not yet been achieved. Kumar et al. proposed that CAFs block the antitumor effects of JNJ-40346527 through the secretion of CXCL1, which induces the infiltration of polymorphonuclear myeloid-derived suppressor cells (PMN-MDSC) within the tumor. Indeed, combined use of the CSF1R inhibitor and a CXCR2 inhibitor abrogated JNJ-40346527-induced PMN-MDSC accumulation and sustained TAM depletion, thereby significantly reducing tumor growth [110]. Anti-PD-1/PD-L1 monoclonal antibodies are currently widely used in NSCLC treatment, but certain patients still exhibit poor responsiveness to immunotherapy. The regulatory effects of CAFs on PD-1 have been previously elucidated in detail. Mounting evidence suggests that the combined use of CAF-targeting therapy and anti-PD-L1 agents can significantly increase patient survival [111,112]. The heterogeneity of antigens on the surface of tumor cells leads to great challenges in the application of antigen-specific CAR-T cells in solid tumors. In addition, the immunosuppressive properties of solid tumor microenvironment may be a key factor interfering with the infiltration of CAR-T cells. Therefore, targeting FAP+CAFs is an appropriate strategy to enhance current immunotherapy. Stable expression of FAP on CAFs can effectively avoid tumor escape and targeting CAFs can help improve tumor microenvironment. CAR therapy for FAP has been shown to be effective in slowing tumor progression [113]. However, there are still great challenges regarding FAP-CAR-T. FAP is not specifically expressed on CAFs, and its expression on bone marrow stromal cells is low. Therefore, targeting FAP-CAR T cells can cause off-target injury to bone marrow stromal cells, inducing significant cachexia and lethal bone toxicities. [114]. We believe that CAF-targeting CAR T cells are promising but identifying CAF-specific markers to reduce off-target damage is necessary.

### 3.4. CAF-Targeted Therapies

As discussed earlier, CAFs can promote tumor formation, growth, invasion, metastasis, and resistance to drugs. In addition to influencing the interactions of CAFs with tumor cells or other mesenchymal cells to achieve anticancer effects, research should also be focused on the direct targeting of CAFs.

#### 3.4.1. FAP

The dipeptidyl peptidase 4 (DPP4) protein family consists of four enzymes, namely DDP4, FAP, DDP8, and DDP9. In particular, DPP enzymatic activity enables the hydrolysis of a prolyl bond two amino acids away from the N-terminus of a protein, making it a target of interest among researchers. FAP is a cell-surface serine protease that possesses not only DPP activity identical to that of DPP4, but also endopeptidase activity towards X-Gly-Pro-Y motifs [115]. Therefore, FAP has attracted widespread interest in studies on cancer and fibrosis. Considering the significant upregulation of FAP in CAFs, targeting CAFs through FAP may represent a potential therapeutic strategy for the treatment of NSCLC. Currently, FAP targeting has been mainly achieved through the use of FAP inhibitors, FAP-activated prodrugs, anti-FAP antibody-drug conjugates, and FAP-specific chimeric antigen receptor T cells (FAP-CAR-T cells).

Preparation of specific FAP inhibitors is a challenging task as FAP possesses both DPP and endopeptidase activities. Therefore, broad-spectrum inhibitors targeted towards the DPP4 family are also worthy of scientific pursuit. Nonetheless, a DPP4 inhibitor was shown to reduce the total number of liver nodules and increase intrahepatic inflammatory cell infiltration in a mouse model of liver cancer [116]. However, a phase II clinical trial of a broad-spectrum DPP inhibitor (talabostat) failed to demonstrate that DDP inhibition could enhance the clinical activity of docetaxel in NSCLC patients [117]. Currently, a considerable number of challenges exist regarding FAP inhibitor development, with the main limitation possibly being off-target effects caused by the dual enzymatic activities of FAP.

FAP-activated prodrugs are initially rendered into non-toxic compounds by FAP through its specific protease activity and are subsequently activated into their cytotoxic form in the TME; this two-step approach enables the reduction of cytotoxic effects of the drugs. Sun et al. developed a prodrug (PCP@R848/DOX) that could be specifically hydrolyzed by FAP. In a xenograft model derived from the MC38 cell line, this triple combination of an amphiphilic bifunctional PD-1/PD-L1 peptide antagonist PCP, R848, and doxorubicin was capable of blocking PD-1/PD-L1 signaling and activating innate immune cells, which in turn enabled significant improvement of the immunosuppressive microenvironment and triggered immunogenic cell death [118]. Prodrugs have also been prepared from other potent cytotoxins, such as thapsigargin and doxorubicin, for reducing systemic toxic side effects, and the use of these prodrugs in xenograft models of multiple types of cancer has led to the achievement of significant antitumor effects.

A combination of anti-FAP antibodies with drugs to form anti-FAB antibody-drug conjugates is another attractive treatment strategy for therapeutically targeting the TME. Sum et al. developed a bispecific FAP-CD40 double antibody that only promoted the activation of CD40 when FAP was present [119]. Results of an in vivo experiment indicated that the FAP-CD40 antibody enhanced intratumoral T cell inflammation, inhibited tumor growth, and attenuated the systemic toxicity of non-targeted CD40 agonists [120], thereby representing a potential strategy for enhancing the effects of CD40 foe anticancer immunotherapy. In another study, a FAP-targeted 4-1BB agonist (FAP-4-1BBL) induced 4-1BB agonistic T cell activation, strengthened T cell effector function in tumors, and reduced the liver toxicity of the agonistic anti-human 4-1BB monoclonal antibody, leading to increased cancer cell apoptosis. Noteworthily, FAP-4-BBL was also capable of mediating tumor cell apoptosis when FAP expression was low [121].

CAR-T cells can recognize target cells in a non-major histocompatibility complex-dependent manner and are mostly used for the treatment of hematological malignancies. In the process of developing CAR-T cells for targeting solid tumors, in addition to the difficulty in finding tumor-specific antigens, the immunosuppressive microenvironment also represents significant challenges to researchers. Given the ability of CAFs to stably express FAP, studies on FAP-CAR-T cells have gained increasing research attention. In a ^51^Cr release assay, FAP-CAR-T cells exhibited significant cytotoxicity towards A549 cells overexpressing FAP. In addition, the FAP-CAR-T cells significantly upregulated the expression of IFN-γ and the tumor necrosis factor-α, thereby killing the FAP^+^ target cells [113]. It has also been reported that depletion of FAP^+^ cells reduces tumor growth in an immune-dependent manner, which makes it potentially applicable for combined use with immune checkpoint inhibitors. Another study on the efficacy of FAP vaccines has also demonstrated the ability of FAP to induce the immune responses of T cells [122].

#### 3.4.2. Leucine-Rich Repeat Containing 15 (LRRC15)

LRRC15 is expressed by the CAFs of lung cancer and multiple types of solid tumors [17,123]. ABBV-085 is an antibody-drug conjugate consisting of a LRRC15-specific monoclonal antibody conjugated to the potent cell-permeable anti-mitotic agent monomethyl auristatin E (MMAE). Upon localization to LRRC15-rich stroma, the active load of the cell-permeable MMAE is dispersed to neighboring cancer cells, where its therapeutic effects can be achieved by promoting mitotic arrest in tumor cells [124].

### 3.5. Targeting Fibrosis in Lung Cancer

Concomitant occurrence of stromal fibrosis in lung cancer has been well demonstrated, and researchers have gradually realized that targeted fibrosis treatment in lung cancer may serve as a promising therapeutic strategy. Pirfenidone is a widely recognized antifibrotic agent that was shown to effectively activate the apoptotic pathways of CAFs of NSCLC, and its combined use with cisplatin exerts strong inhibitory effects on cancer cells and CAFs [125]. Nintedanib is a multi-kinase inhibitor that suppresses CAF activation, reduces the secretion of various cytokines and growth factors that stimulate cancer cell growth and migration, and provides powerful antifibrotic effects. Thus, nintedanib may contribute to the destruction of the physical barriers of drug transport, which is illustrated by the enhancement of therapeutic effects when used in combination with docetaxel [126]. Interestingly, it is believed that the poor responsiveness to nintedanib in squamous cell carcinoma is attributed to significant differences in integrin signals as compared with adenocarcinoma [127]. Moreover, reduced fibrosis, as well as the reduced nintedanib response of LUSC-related CAFs as compared with LUAD-related CAFs, were found to be associated with the increased promoter methylation of the profibrotic TGF-β transcription factor SMAD3. This event results in a compensatory increase of TGFβ1/SMAD2 activation, thereby reducing the LUSC fibrotic phenotype [128].

## 4. Conclusions

Research on the roles of CAFs in NSCLC treatment and drug resistance has rapidly progressed in recent years. Given the vital role of CAFs in tumor onset and progression, using CAFs as novel therapeutic targets for NSCLC treatment has gathered significant attention. By targeting CAFs, the effectiveness of current treatments can be enhanced, and prevention and treatment of tumor metastasis can be achieved, thereby paving the way for tackling cancer progression. However, given the dynamic and complex behavior of the TME, our understanding of the interactions of CAFs with tumor cells, immune cells, and other stromal components remain largely insufficient. Another barrier to CAF targeting is the current lack of clinically recognized specific markers. Therefore, further in-depth investigations on the roles of CAFs are still required to elucidate the various molecular components of the TME and discover specific CAF markers, so as to provide a solid theoretical foundation for NSCLC treatment.

## Figures and Tables

**Figure 1 pharmaceuticals-15-01411-f001:**
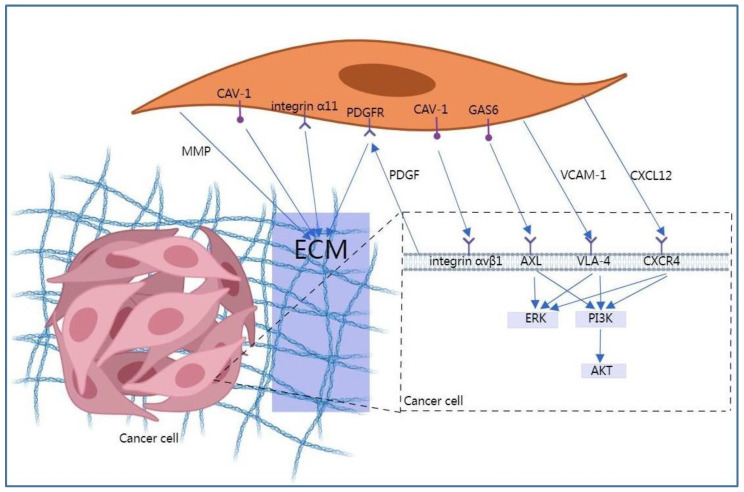
Cancer-associated fibroblasts (CAFs) mediate NSCLC metastasis and invasion. CAV-1, Caveolin-1; CXCL12, C-X-C motif chemokine ligand 12; ECM, extracellular matrix; GAS6, Growth arrest-specific 6; MMP, Matrix metalloproteinase; PDGF, Platelet-derived growth factor; PDGFR, Platelet-derived growth factor receptor; VCAM-1, Vascular cell adhesion molecule-1.

**Figure 2 pharmaceuticals-15-01411-f002:**
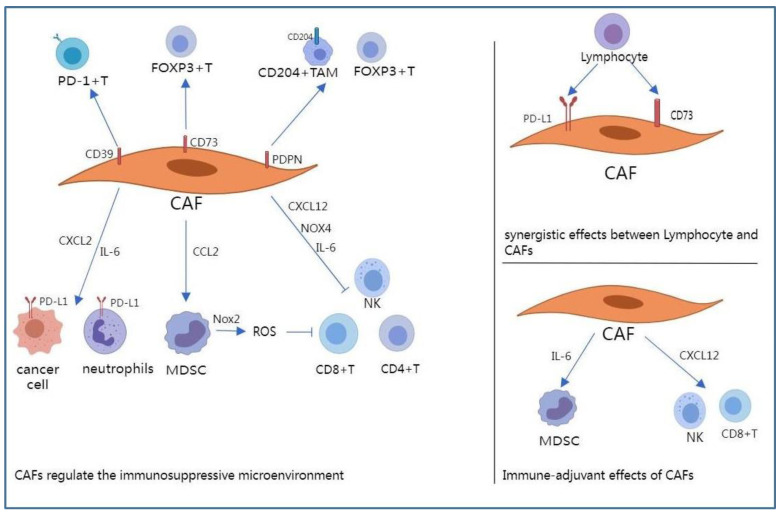
Cancer-associated fibroblasts (CAFs) regulate the immune microenvironment of NSCLC mediate NSCLC. MDSC, myeloid-derived suppressor cell; NK, natural killer cell; TAM, tumor-associated macrophages.

**Table 1 pharmaceuticals-15-01411-t001:** Cancer-associated fibroblasts mediate resistance to tyrosine kinase inhibitors (TKIs).

Agent	Cell Lines	Mode of Action	Inhibit	Ref.
HGF	A549, HI1017, and EBC1	Bypass pathway: HGF→MET→PI3K→AKT	Anti-HGF antibody and MET-TKIs	[91]
	PC9 and HCC827		Anti-HGF antibody and HGF antagonist	[92]
	HCC827 and H1975		MET-TKIs	[93]
FGF	—	Promotes MET	FGFR inhibitor: BGJ398 and others	[94]
	H1975and others	Bypass pathway: FGFR1→STAT3; p42/p44; AKT	FGFR inhibitor: AZD4547	[95]
	HCC827	Bypass pathway: FGF→FGFR1→PI3K→AKT FGF2→FGFR1→MEK→ERK	FGFR inhibitor: PD173074	[96]
	PC9, HCC827, and others	Bypass pathway: FGF→FGFR3→MEK→ERK	FGFR inhibitor (BGJ398)	[97]
GAS6	—	Bypass pathway: GAS6→AXL	AXL inhibitor: SGI-7079	[42]
OSM	—	Bypass pathway: OSM→OSMR↔JAK→STAT3	JAK1 inhibitor: filgotinib	[40]
MEK→ERK	—	MEK→ERK→miR-21→PDCD4↓	MEK inhibitor: trametinib	[98]

**Table 2 pharmaceuticals-15-01411-t002:** Cancer-associated fibroblast-mediated drug resistance in chemotherapy.

Agent	Cell Lines	Mode of Action	Inhibit	Ref.
IL-6	A549 and NCI-H358	IL-6→IL-6R	—	[100]
	A549, SW900, and H520	IL-6→IL-6R→JAK→STAT3	T21	[101]
IL-11	A549 and H1975	IL-11→IL-11R→JAK→STAT3	—	[102]
IGF	A549	IGF2→IGF-1R→AKT/SOX2→ABCB1	IGF-1R inhibitor: OSI-906	[103]
Collagen type I	H460 and H1299	Collagen→integrin→Src	Src inhibitor: dasatinib	[104]
CCL5	A549 and H1299	CCL5→CCR5→lncRNAHOTAIR	CCR5 antagonist: Met-RANTES	[80]
GAS6	H1299	GAS6→AXL	AXL inhibitor: TP-0903	[41]
GGT5	A549 and LAC1	GGT5→GSH→ROS↓	GGT inhibitor: GGsTop	[105]
ANXA3	ANXA3	ANXA3→JNK→survivin	JNK inhibitor: SP600125	[106]

## Data Availability

The study did not report any data.

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
