# Peer review of "Research Progress on Therapeutic Targeting of Cancer-Associated Fibroblasts to Tackle Treatment-Resistant NSCLC"

_pharmaceuticals, 2022, doi:10.3390/ph15111411_

Round 1

Reviewer 1 Report

Li et al. provide a review on the biomolecular implication of CAFs in the setting of NSCLC with emphasis on their ability to promote tumor progression and overall resistance to a multitude of therapeutic modalities. The review is very comprehensive and scientifically accurate, but also extremely long and difficult to read. In this context, the work might appear unappealing to a large portion of the readership looking to find quick snapshots on the topic. A possible solution would be to condense the lengthy text narrative using more figures with diagrams.

Additional minor point of criticism are appended below.

Title – The perceived message of the current title is misleading. It appears to convey the notion of CAFs as some sort of cellular therapy while CAFs represent the therapeutic target. In that context, consider specifying that CAFs are the actual therapeutic target in the title. One possibility would be to replace “use” with “targeting”.

Introduction, line 60 – Consider removing “various cytokines” as structural components of the TME are being described and cytokines do not belong to this category.

Introduction, lines 74-77 – This definition of CAFs does not apply for some types of mesenchymal tumors with spindle cell morphology, such as undifferentiated sarcomas, where neoplastic cells do not have endothelial, leukocytic, or epithelial histogenesis.

Introduction, line 92 – Consider replacing the term “non-precise” with “unspecific”.

Figure 1 – the abbreviation CDT is not explained.

2.2. CAFs regulate the immune microenvironment of NSCLC, line 202-210 – The introduction to this section is not completely clear and should be better elaborated/expanded.

2.2.1. NADPH oxidases, lines 212-215 – It is unclear why the increase of MDSC should promote tumor suppression. Tumors with high levels of infiltration by MDSCs have been associated with poor patient outcome and resistance to therapies. A better explanation with references should be provided.

2.2.7. Immune-adjuvant effects of CAFs, lines 290-291 – This aspect should be better explained as it contradicts what has been reported previously on IL-6 and CXCL12.

2.3.2.4. Necrosis – Necrosis is a rather nebulous and nonspecific term that can be applied to diverse types of accidental cell death or regulated cell death. The authors should specify what cell death modality is considered in this work.

3.3. CAFs regulate responsiveness to immunotherapy – Importantly, this section should include the negative effect of CAFs on cellular immunotherapies including CAR T cells. These latter have been demonstrated to be very effective against liquid malignancies but their efficacy in solid tumors is still limited. In this context the TME, including CAFs, is primarily responsible for preventing the distribution and cytotoxic effects of CAR T cells.

Author Response

We appreciate the reviewers very much for their constructive comments and suggestions, which helped a lot to improve our manuscript. We have studied comments carefully and have made correction which we hope meet with approval.

Comment 1: A possible solution would be to condense the lengthy text narrative using more figures with diagrams.

Response 1: Thanks for the suggestion, Figure2 has been added.

Comment 2: Title – The perceived message of the current title is misleading. It appears to convey the notion of CAFs as some sort of cellular therapy while CAFs represent the therapeutic target. In that context, consider specifying that CAFs are the actual therapeutic target in the title. One possibility would be to replace “use” with “targeting”.

Response 2: It has been recorrected in the revised manuscript.

Comment 3: Introduction, line 60 – Consider removing “various cytokines” as structural components of the TME are being described and cytokines do not belong to this category.

Response 3: It has been recorrected in the revised manuscript.

Comment 4: Introduction, lines 74-77 –This definition of CAFs does not apply for some types of mesenchymal tumors with spindle cell morphology, such as undifferentiated sarcomas, where neoplastic cells do not have endothelial, leukocytic, or epithelial histogenesis.

Response 4: Thanks for the suggestion. After reviewing relevant studies, CAF is the mesenchymal part of malignant tumors derived from epithelial tissues. Thus, this review did not involve mesenchymal tumors.

Comment 5: Introduction, line 92 –Consider replacing the term “non-precise” with “unspecific”.

Response 5: It has been recorrected in the revised manuscript.

Comment 6: Figure 1 – the abbreviation CDT is not explained.

Response 6: Figure 1 has been changed. And we explained the abbreviations in the new figure.

Comment 7: 2.2. CAFs regulate the immune microenvironment of NSCLC, line 202-210 – The introduction to this section is not completely clear and should be better elaborated/expanded.

Response 7: Thanks for the suggestion, the content about CAFs regulating the immune microenvironment of NSCLC has been expanded in the lines 214-224 of the revised manuscript.

Comment 8: 2.2.1. NADPH oxidases, lines 212-215 – It is unclear why the increase of MDSC should promote tumor suppression. Tumors with high levels of infiltration by MDSCs have been associated with poor patient outcome and resistance to therapies. A better explanation with references should be provided.

Response 8: Thanks for the comment. We are also deeply aware of the defects of the original text, so it has been recorrected in lines 232-245 of the revised manuscript.

Comment 9: 2.2.7. Immune-adjuvant effects of CAFs, lines 290-291 – This aspect should be better explained as it contradicts what has been reported previously on IL-6 and CXCL12.

Response 9: Thanks for the comment, we explained the immune-adjuvant effects of CAFs from three aspects in lines 308-331 of revised manuscripts.

Comment 10: 2.3.2.4. Necrosis – Necrosis is a rather nebulous and nonspecific term that can be applied to diverse types of accidental cell death or regulated cell death. The authors should specify what cell death modality is considered in this work.

Response 10: Thanks for the comment, in pathology, Necrosis can be the outcome of any pattern of cell death. We realize that subheading necrosis in the PCD is unreasonable and strongly misleading. We have deleted this part in the revised manuscript.

Comment 11: 3.3. CAFs regulate responsiveness to immunotherapy – Importantly, this section should include the negative effect of CAFs on cellular immunotherapies including CAR T cells. These latter have been demonstrated to be very effective against liquid malignancies but their efficacy in solid tumors is still limited. In this context the TME, including CAFs, is primarily responsible for preventing the distribution and cytotoxic effects of CAR T cells.

Response 11: Thanks for the comment. It has been added in lines 602-614 of the revised manuscript.

Reviewer 2 Report

Well written review and this article will be of interest to the pharmaceutics scientific community for an overview of strategies to shift the TME 

The authors approached the topic with a broad range of effects of CAFS on the TME and some sections could be elaborated with more detail

For example line 85 what is the significance of the three subtypes with respects to resistance?

Table 1 could use some more detail not clear as shown what the influence/factors produce by  the CAF influence the MDSC and TAM

VCAM1 section is underdeveloped what about effects on survival pathways and immune subsets rich in secreted VCAM1?

Expanding the information on any clinical information relating CAFS to poor outcome would enhance the relevance of Table 1 an 2 which are cell line based.  Adding a table which has documented clinical significance would strengthen the review

Author Response

We appreciate the reviewers very much for their constructive comments and suggestions, which helped a lot to improve our manuscript. We have studied comments carefully and have made correction which we hope meet with approval.

Comment 1: For example, line 85 what is the significance of the three subtypes with respects to resistance?

Response1: Thanks for the comment. Three subtypes directly impact clinical anticancer response. It has been expanded in the revised manuscript, lines 85-93.

Comment 2: Table 1 could use some more detail, not clear as shown what the influence/factors produce by the CAF influence the MDSC and TAM

Response2: Thanks for the comment. The figure has been changed. And the factors produced by the CAF influence the MDSC and TAM are shown in figure2.

Comment 3: VCAM1 section is underdeveloped. what about effects on survival pathways and immune subsets rich in secreted VCAM1?

Response3: Thanks for the comment, VCAM-1 section have been modified and elaborated in the revised manuscript, lines127-140.

Comment 4: Expanding the information on any clinical information relating CAFS to poor outcome would enhance the relevance of Table 1 and 2 which are cell line based. Adding a table which has documented clinical significance would strengthen the review.

Response4: As for the clinical information of table1 and table2, only one research(ref41) involves a clinical cohort, other studies were based on cell lines and xenograft models. So clinical information is difficult to add in the tables. In reference [41], a total of 69 patients were analyzed. Tumor Axl and stromal Gas6 expression were determined by immunohistochemistry. Encouragingly, stromal Gas6 expressed and secreted by CAFs tended to be increased after chemotherapy, and the increase of Gas6 was associated with distant metastasis and local recurrence of patients.
